# Correlation between Previous Antibiotic Exposure and COVID-19 Severity. A Population-Based Cohort Study

**DOI:** 10.3390/antibiotics10111364

**Published:** 2021-11-08

**Authors:** Carl Llor, Dan Ouchi, Maria Giner-Soriano, Ana García-Sangenís, Lars Bjerrum, Rosa Morros

**Affiliations:** 1Fundació Institut Universitari per la Recerca a l’Atenció Primària de Salut Jordi Gol i Gurina (IDIAPJGol), 08007 Barcelona, Spain; douchi@idiapjgol.info (D.O.); mginer@idiapjgol.info (M.G.-S.); agarcia@idiapjgol.org (A.G.-S.); rmorros@idiapjgol.org (R.M.); 2Department of Public Health, General Practice, University of Southern Denmark, 5000 Odense, Denmark; 3Universitat Autònoma de Barcelona, Bellaterra, 08193 Cerdanyola del Vallès, Spain; 4Section and Research Unit of General Practice, Department of Public Health, University of Copenhagen, 1014 Copenhagen, Denmark; LBjerrum@sund.ku.dk; 5Departament de Farmacologia, Terapèutica i Toxicologia, Universitat Autònoma de Barcelona, Bellaterra, 08193 Cerdanyola del Vallès, Spain; 6Plataforma SCReN, IICEC IDIAP Jordi Gol, 08007 Barcelona, Spain

**Keywords:** drug resistance, microbial, primary health care, COVID-19, anti-bacterial agents

## Abstract

We examined the correlation between previous antibiotic exposure and COVID-19 severity using a population-based observational matched cohort study with patient level data obtained for more than 5.8 million people registered in SIDIAP in Catalonia, Spain. We included all patients newly diagnosed with COVID-19 from March to June 2020 and identified all their antibiotic prescriptions in the previous two years. We used a composite severity endpoint, including pneumonia, hospital admission and death due to COVID-19. We examined the influence of high antibiotic exposure (>4 regimens), exposure to highest priority critically important antimicrobials (HPCIA) and recent exposure. Potential confounders were adjusted by logistic regression. A total of 280,679 patients were diagnosed with COVID-19, 146,656 of whom were exposed to at least one antibiotic course (52.3%) during the preceding two years. A total of 25,222 presented severe COVID-19 infection (9%), and the risk of severity was highest among those exposed to antibiotics (OR 1.12; 95% CI: 1.04–1.21). Among all individuals exposed to antibiotics, high, recent and exposure to HPCIAs were correlated with increased COVID severity (OR 1.19; 95% CI: 1.14–1.26; 1.41; 95% CI: 1.36–1.46; and 1.35; 95% CI: 1.30–1.40, respectively). Our findings confirm a significant correlation between previous antibiotic exposure and increased severity of COVID-19 disease.

## 1. Introduction

Coronavirus disease 2019 (COVID-19), caused by the SARS-CoV-2 coronavirus strain, has caused over one hundred million infections and over two million deaths to date [1]. In about 80% of cases, the virus resides in the upper respiratory tract leading to an innate immune response that is mild and requires conservative symptomatic therapy. The remaining 20% of cases experience a much more severe form of the disease [2]. Disease severity has shown to be associated with older age and the presence of comorbidities, specifically including cardiovascular disease, diabetes, respiratory disease and smoking [3]. However, the reasons why some patients deteriorate are not yet fully known, and despite the fact that older individuals are far more likely to become critically ill or die from COVID-19, stark differences in death rates have been observed across countries [4]. Bacterial superinfections in these patients have been postulated and this explains why three quarters of patients with COVID-19, especially during the first wave of the pandemic, received antibiotics empirically [5,6]. Recent studies have also stressed the importance of the gut microbiome influencing the severity of the coronavirus infection. The composition of the gut microbiome might influence not only the severity of COVID-19, but also the magnitude of immune system response to infection [7].

Use of systemic antibiotics, preferentially broad-spectrum antibacterials, has been observed to have several impacts on gut health, including reducing microbial diversity in the gut, reducing protective species such as *Bifidobacterium* spp. and promoting the colonization of opportunistic pathogens such as *Clostridium difficile* that can cause antibiotic-associated diarrhea [8,9,10]. Antibiotic consumption results in an increased isolation of resistant germs and countries with high antibiotic consumption rank at the highest levels of antimicrobial resistance [11]. The problem of resistance not only involves the community, but also affects the individual [12]. Increased antimicrobial resistance is also the cause of severe infections, complications, longer hospital stays and increased mortality [13]. A relationship between the isolation of resistant germs and mortality among patients with COVID-19 infection is therefore very likely.

The interplay between bacteria, viruses and host physiology is complex, and we still have much to learn. Despite this, an increasing body of evidence is beginning to reveal how antibiotic exposure appears to impair antiviral immunity. Some papers carried out in rodents have found how antibiotic exposure among pregnant mice causes substantial alterations to theirs and their offspring′s gastrointestinal microbiota and increased mortality following viral infection [14]. The evidence in humans is, however, scarce. In a recent study, Zhou et al. showed that perinatal antibiotic exposure for preventing group B streptococcus infection in newborns induced microbiota dysbiosis in maternal vaginal and neonatal gut environments and was associated with an increased risk of the occurrence of early-onset sepsis among the latter [15]. Another study showed how dysbiosis within the vaginal microbiota caused by oral antibiotic treatment results in severe impairment of antiviral protection against herpes simplex virus type 2 infection [16]. Probiotics, however, would have an opposite role and some selected probiotics have been reported to increase natural killer cell activity and cytotoxic activity [17]. We hypothesized that previous antibiotic use is correlated with an increased severity of COVID-19 disease. We hypothesized that the risk was greatest in patients with high antibiotic exposure (>four antibiotic regimens), recent exposure to antibiotics (previous two months) and exposure to the so-called broad-spectrum highest priority critically important antimicrobials (HPCIA). We therefore took advantage of routinely collected primary care data to study the association between the exposure to antibiotics in the previous two years and disease severity in patients diagnosed with COVID-19 in the general population of Catalonia.

## 2. Results

### 2.1. Study Population

A total of 280,679 patients with a mean age of 46.3 years were diagnosed with confirmed and suspected COVID-19 infection during the study period. A total of 146,656 patients were exposed to at least one antibiotic course in the previous two years (52.3%), with 14,915 patients taking five or more antibiotic courses. Patients exposed to antibiotics were significantly older than those not exposed (48.1 vs. 44.5 years). Smoking status, body mass index, percentage of obese patients and deprivation status were greater for those exposed to antibiotics (Table 1). As shown in Table 1, the presence of the different comorbidities of interest as well as the concomitant use of non-steroidal anti-inflammatory drugs, antithrombotic medication, corticosteroids and heparin were significantly higher among patients exposed to antibiotics. Penicillins accounted for 181,815 antibiotics dispensed (47.9%) (Appendix A). In total, 59,176 had taken their last course less than two months prior to COVID-19 infection (40.4%). A total of 47,477 patients had taken at least one HPCIA during the 2-year period.

### 2.2. The Correlation between Previous Use of Antibiotics and COVID-19 Severity

A total of 25,222 patients presented the primary composite outcome of death, hospitalization and/or pneumonia related to COVID-19 infection (9% of the patients). This percentage of disease severity was significantly higher among patients exposed to antibiotics (n: 15,828, 10.8%) compared to patients not exposed (n: 9394, 7%) (Figure 1). In the multivariable analysis, we found a significant correlation between previous antibiotic exposure and the composite severity endpoint (OR 1.12; 95% CI: 1.04–1.21) (Table 2). High antibiotic exposure (five or more antibiotic regimens) was associated with an OR of severity of 1.19 (95% CI: 1.14–1.26) compared to medium + low antibiotic regimens. Patients with recent antibiotic exposure (<2 months) had a higher risk of COVID-19 severity than patients with past antibiotic exposure (OR 1.41; 95% CI: 1.36–1.46). Patients with past antibiotic exposure (≥2 months) had no increased risk of COVID-19 severity. Patients exposed to HPCIA had a significantly higher risk of COVID-19 severity than patients exposed to other antibiotics (OR, 1.35; 95% CI: 1.30–1.40). Among individuals who were exposed to antibiotics, those who presented the most severe cases of COVID-19 infection were the patients exposed to quinolones (OR, 1.49; 95% CI: 1.42–1.56) and cephalosporins (OR, 1.45; 95% CI: 1.36–1.55) (Appendix A). Figure 2 shows adjusted ORs of the correlation between different antibiotic exposures and the composite severity endpoint of COVID-19.

Patients with previous exposure to antibiotics had a non-significant increased risk of pneumonia (OR: 1.18; 95% CI: 0.81–1.74), and significant increased risk of hospital admission (OR: 1.14; 95% CI: 1.02–1.27) (Appendix A). The risk of death was significantly higher among individuals taking five or more antibiotic courses in the previous two years compared to those with a lower exposure (OR, 1.44; 95% CI: 1.33–1.55). Recent antibiotic use was associated with an increased probability of pneumonia and hospitalization, but a lower death rate. Notwithstanding, the use of HPCIAs was significantly associated with a higher risk of both pneumonia, hospital admission and death (Appendix A).

## 3. Discussion

Catalonia was hard-hit by the first wave of the COVID-19 pandemic and it constitutes a suitable region to investigate risk factors of COVID-19 severity. Our study assessed the composite endpoint of COVID-19 severity combining pneumonia, hospital admission and/or death in two cohorts of patients—those exposed to antibiotics in the previous two years and those not exposed. The results of our study reveal that there is a correlation between previous antibiotic exposure and severity. Multivariable analysis showed that most of this effect was due to underlying chronic diseases and the concomitant use of other medications, but the effect continued being statistically significant after adjustment for potential confounders. In addition, as hypothesized, this correlation between previous exposure and COVID-19 severity was strongest among individuals with high antibiotic exposure, recent exposure, and exposure to HPCIA.

To our knowledge, this is the first population-based study in our setting to assess the relationship between previous exposure to antibiotics and COVID-19 severity. The strengths of this study are the large number of patients included, complete sociodemographic data and real-world data. Our data are representative of the Catalan general population as stated by other previous studies [18]. This study also has some limitations inherent to electronic database studies, such as incompleteness, potential confounders, non-randomized data, and possible selection biases. Furthermore, the real exposure of antibiotics may vary from obtained prescription data. Odds ratios have been adjusted for potential confounders that might have an influence on COVID-19 severity. Both confirmed and suspected cases of COVID-19 were considered in this study. During the first wave of the COVID-19 pandemics in Catalonia, PCR tests were not routinely conducted in all patients with suspected COVID-19 infection, due to the unavailability of laboratory kits. In this study, we decided to include both patients with a confirmed diagnosis of COVID-19 by PCR and a suspicious diagnosis (hospital registered diagnosis of COVID-19 infection but not confirmed by PCR). The number of cases obtained was comparable to the official statistics of the community.

A modest but significant correlation was observed between antibiotic use and COVID-19 severity. Approximately half of the patients who developed COVID-19 infection between 1 March and 30 June 2020 were previously prescribed at least one antibiotic course, and this allowed two cohorts of a similar number of individuals to be compared.

Microbial communities are scattered all over and inside the human body, i.e., skin, vagina, intestine, and oral cavity. The abundance, diversity, and features of the genes of microorganisms are collectively known as the human microbiome. However, most organisms are harbored by the gastrointestinal tract. Maintaining the variety and balance of the gut microbiota is essential for promoting human health. Alterations in the diversity or structure of the gut microbiota, known as dysbiosis, have been associated with chronic diseases ranging from gastrointestinal inflammatory and metabolic conditions to neurological, cardiovascular, and respiratory illnesses [19]. Gut dysbiosis has recently been suggested to also influence immune system response to COVID-19 and potentially affect the severity and outcome of the disease. By obtaining blood and stool samples and medical records from patients hospitalized for COVID-19 as well as healthy people in the first three months of the pandemic, Yeoh et al. showed that the composition of the microbiome significantly differed between patients with and without the disease [7].

Systemic antibiotics are known to alter the gut microbiome. There is compelling evidence that antibiotic treatment is followed by a significant alteration of gut microbiota composition and a reduction of between one-fourth to one-third of the microbial diversity in the digestive tract [8,9]. This effect has been shown to be greater with broad-spectrum antibiotics. In order to prioritize and protect the most essential antibiotics used in human medicine, the World Health Organization and national governments have created a list of HPCIAs, reflecting the great importance these antibiotics have for treatment of multidrug-resistant infections. These antimicrobials with a broader antimicrobial spectrum and judged to be essential for human medicine should be restricted not only from use in animal feed but also in human medicine [20].

Recent studies on the long-term effects of antibiotic intake have shown that the microbiota does not exhibit complete resilience two months after treatment cessation [21]. In our study, patients who had taken their last antibiotic course in the previous two months presented significant increased COVID-19 severity. It has been shown that resistant organisms can be recovered from individuals who had been exposed to antibiotics up to one year before [12]. The maintenance of a possible increase in microbial load associated with a decrease in diversity suggests that the eviction of microorganisms sensitive to antibiotics provides space for resistant strains to overgrow and dominate the niche. This could explain why patients with recurrent antibiotic exposure had a higher risk of severe COVID-19 infection. 

During 2020, the COVID-19 pandemic, an unprecedented global public health crisis, profoundly altered the context for antimicrobial stewardship, mainly in primary care. After indiscriminate antibiotic prescribing during the first wave of the pandemic, the WHO issued guidance on the clinical management of COVID-19, recommending that antibiotics should not be prescribed for the prevention and treatment of mild COVID-19 infection. The NICE guideline subsequently suggested that COVID-19 patients presenting with pneumonia symptoms are more likely to have a viral origin than a community-acquired bacterial pneumonia, and thus, antibiotic prescriptions should be offered only where bacterial infections are suspected [22]. A subsequent meta-analysis confirmed this, as only 3.5% of all COVID-19 patients present with bacterial co-infection [23]. Therefore, there is no evidence that common infections such as respiratory and urinary tract infections should be managed differently during the pandemic, since inappropriate use of antibiotics to treat viral infections and indiscriminate use of broad-spectrum antibacterials may lead to resistance [24]. In addition, this practice is associated with deleterious effects on the microbiome and may pose a risk for subsequent severe viral infections, as shown in the current study. The ongoing overall crisis of antimicrobial resistance must not be neglected and advocacy for antimicrobial stewardship must continue during the pandemic and the post-pandemic era.

## 4. Methods

### 4.1. Study Design, Setting and Participants

We conducted a population-based retrospective cohort study including adult patients diagnosed with COVID-19 registered as confirmed (by polymerase chain reaction (PCR)) or as suspicious (not confirmed by PCR but with the diagnosis registered) in the primary healthcare records in Catalonia, Spain, from the onset of the pandemic, on 1 March 2020 to 30 June 2020. Patients were grouped into two cohorts according to their previous exposure to antibiotics within a period of two years prior to COVID-19 diagnosis. Patients who were residents in nursing homes (long-term facilities) were excluded from the study.

### 4.2. Data Collection

The study data source was the Information System for Research in Primary Care (SIDIAP; www.sidiap.org (accessed on 4 November 2021)) database [25], which captures the clinical information of approximately 5.8 million people in Catalonia, Spain (around 80% of the Catalan population and is representative in geography, age, and sex). This information is pseudonymized, and originates from different data sources: (a) ECAP (electronic health records in primary care of the Catalan Health Institute), including socio-demographic characteristics, comorbidities registered as International Classification of Disease (ICD)-10 codes [26], specialist referrals, clinical parameters, toxic habits, sickness leave, date of death, laboratory test data, and drug prescriptions issued in primary healthcare, registered in the Anatomical Therapeutic Chemical (ATC) classification system [27]; (b) pharmacy invoice data corresponding to the primary healthcare drug prescriptions; (c) database of diagnoses at hospital discharge [28]; and (d) COVID-19 data from the Catalan Agency of Health Quality and Evaluation (AQuAS) [29]. The codes considered in this study are summarized in Appendix A. The SIDIAP database has been extensively used for national and international epidemiologic and pharmacoepidemiologic studies and has been validated in primary care [30].

### 4.3. Drug Exposure

At a pharmacological group level, prescriptions with intervals of less than two weeks were combined into a single treatment episode. Patients were classified as exposed to systemic antibiotics (J01 of the ATC classification system, Antibacterials for systemic use) if they had at least one treatment episode with any antibiotic during the previous two years prior to COVID-19 diagnosis and were grouped according to the level of exposure: low, defined as exposure to one or two antibiotic courses; medium, with three or four courses; and high, if patients were exposed to five or more antibiotic courses in the previous two years. Recent exposure was defined by less than two months prior to COVID-19 infection diagnosis. Antimicrobials considered as HPCIA were quinolones, third- and fourth-generation cephalosporins, macrolides, ketolides, and glycopeptides [31].

### 4.4. Variables and Outcomes

The index date for all individuals was the date of COVID-19 infection. At this date, the variables captured were as follows: gender; age; geographical area; mortality in small Spanish areas and socioeconomic and environmental inequalities (MEDEA) socioeconomic deprivation score including five discrete values [32]; obesity, defined as a body mass index greater than 30 kg/m^2^; smoking habit; comorbidities of interest; and concomitant use of medications of interest. 

The variables assessed in each cohort during follow-up were the diagnosis of pneumonia, hospital admission and mortality—all related to COVID-19 infection. The primary outcome was a composite endpoint of severity composed by pneumonia, hospitalization and/or death due to COVID-19. The risk of these events was analyzed comparing patients exposed to antibiotics with those non-exposed and according to the level of exposure to antibiotics: high vs. low exposure, recent vs. old exposure and exposure to HPCIAs vs. other antibiotics. We assumed that in individuals with no record of any of the variables of interest, there were no existing severity endpoints, health problems, drug exposure or smoking habits.

### 4.5. Statistical Analysis

Quantitative variables were described as mean and standard deviation, whereas categorical variables were described as the proportion over the exposed and non-exposed individuals. Univariable tests reported in the descriptive table were obtained from the Student’s *t*-test and Chi-square test as appropriate. For the primary outcome, marginal structural models (MSM) were fitted to estimate causal effects by correcting for confounding. We estimated inverse probability weights (IPWs) based on the propensity score using age, gender, and the socioeconomic deprivation score. If needed, weights were truncated to the 5th percentile. IPWs were used in the MSM to estimate odds ratio (OR) and confidence intervals (CI) comparing the prevalence of each outcome among individuals exposed to antibiotics to those not exposed to antimicrobials. When evaluating the correlation of previous antibiotic exposure with the different outcomes, patients were counted only once and allocated to the worst outcome (death > hospitalization > pneumonia). Variable selection was performed using the Akaike information criterion-based stepwise procedure, and we used the Wald test of the fitted coefficient to determine whether the log-odds was significantly different from zero at a 0.05 level. All analyses were performed using R software (v3.6.3).

## 5. Conclusions

We observed a significant correlation between previous antibiotic exposure and the risk of increased COVID-19 disease severity. This association was even greater in patients who were more intensively exposed and in those who had more recently taken an antibiotic course.

## Figures and Tables

**Figure 1 antibiotics-10-01364-f001:**
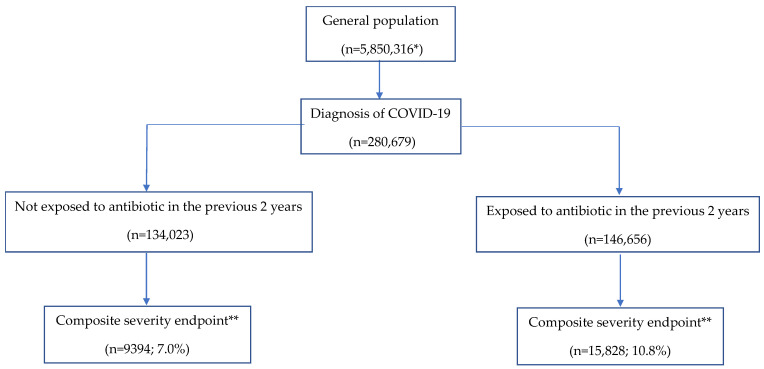
Flowchart of the study participants. * Up to 30 June 2020, ** Includes pneumonia, hospitalization and/or death due to COVID-19.

**Figure 2 antibiotics-10-01364-f002:**
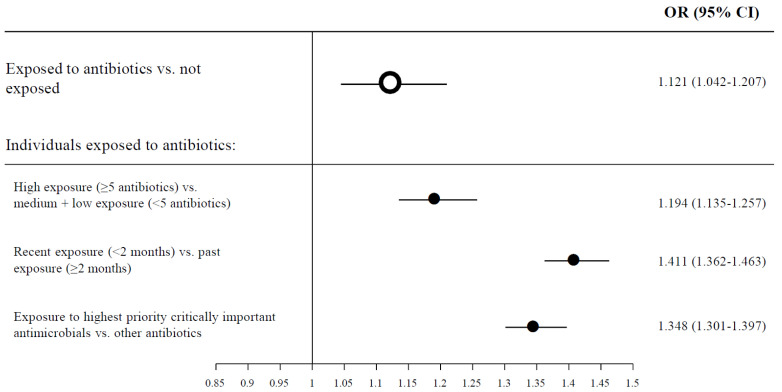
Forest plot showing adjusted odds ratios of the association between different antibiotic exposures and the composite severity endpoint of COVID-19.

**Table 1 antibiotics-10-01364-t001:** Distribution of patients with COVID-19 by exposure to antibiotics in the previous two years.

	Overall (n: 280,679)	Non-Exposed to Antibiotics (n: 134,023)	Exposed to Antibiotics (n: 146,656)	*p*-Value
**Sociodemographic Data ***				
Gender				<0.0001
Female, n (%)	153,034 (54.5)	67,039 (50.0)	85,995 (58.6)	
Male, n (%)	127,645 (45.5)	66,984 (50.0)	60,661 (41.4)	
Age, mean (SD)	46.3 (20.4)	44.5 (19.0)	48.0 (21.5)	<0.0001
Age (categorical)				<0.0001
<60 yr., n (%)	215,699 (76.8)	109,979 (82.1)	105,720 (72.1)	
≥60 yr., n (%)	64,980 (23.2)	24,044 (17.9)	40,936 (27.9)	
Deprivation index score (%)				<0.0001
Unknown, n (%)	71,311 (25.4)	33,967 (25.3)	37,344 (25.5)	
Urban 1st quintile (least deprived), n (%)	39,861 (14.2)	21,207 (15.8)	18,654 (12.7)	
Urban 2nd quintile, n (%)	42,795 (15.2)	21,094 (15.7)	21,701 (14.8)	
Urban 3rd quintile, n (%)	42,511 (15.1)	20,497 (15.3)	22,014 (15.0)	
Urban 4th quintile, n (%)	42,402 (15.1)	19,181 (14.3)	23,221 (15.8)	
Urban 5th quintile (most deprived), n (%)	41,799 (14.9)	18,077 (13.5)	23,722 (16.2)	
**Associated Comorbidity and Risk Factors ***				
Smoking habit, n (%)	110,781 (39.5)	47,788 (35.7)	62,993 (43.0)	<0.0001
Obesity, n (%)	75,739 (27.0)	29,882 (22.3)	45,857 (31.3)	<0.001
Ischemic heart disease, n (%)	7706 (2.7)	2452 (1.8)	5254 (3.6)	<0.0001
Diabetes mellitus, n (%)	23,604 (8.4)	8167 (6.1)	15,437 (10.5)	<0.0001
High blood pressure, n (%)	57,773 (20.6)	21,828 (16.3)	35,945 (24.5)	<0.0001
Heart failure, n (%)	5256 (1.9)	1213 (0.9)	4043 (2.8)	<0.0001
Chronic kidney disease, n (%)	11,915 (4.2)	3467 (2.6)	8448 (5.8)	<0.0001
Respiratory disease, n (%)	45,931 (16.4)	14,060 (10.5)	31,871 (21.7)	<0.0001
Thromboembolism, n (%)	896 (0.3)	260 (0.2)	636 (0.4)	<0.0001
**Concomitant Medication ***				
NSAIDs, n (%)	66,764 (23.8)	22,092 (16.5)	44,672 (30.5)	<0.0001
Antithrombotic medication, n (%)	15,150 (5.4)	5052 (3.8)	10,098 (6.9)	<0.0001
Corticosteroids, n (%)	14,663 (5.2)	3765 (2.8)	10,898 (7.4)	<0.0001
Low molecular weight heparin, n (%)	2122 (0.8)	666 (0.5)	1456 (1.0)	<0.0001
**Antibiotic Exposure**				
Antibiotic exposure intensity ^†^				-
None, n (%)	134,023 (47.7)	134,023 (100.0)	-	
Low (1–2 prescriptions), n (%)	104,873 (37.4)	-	104,873 (71.5)	
Medium (3–4 prescriptions), n (%)	26,868 (9.6)	-	26,868 (18.3)	
High (≥5 prescriptions), n (%)	14,915 (5.3)	-	14,915 (10.2)	
Last antibiotic course taken				
<2 months before COVID-19 infection	59,176 (40.4)	-	59,176 (40.4)	
≥2 months before COVID-19 infection	87,480 (59.6)	-	87,480 (59.6)	
Days to last antibiotic prescription, mean (SD)	198.8 (214.6)	-	198.8 (214.6)	-
Highest priority critically important antimicrobials, n (%)	47,477 (32.4)	-	47,477 (32.4)	-
**COVID-Related Severity Events ^‡^**				
Death, hospitalization and/or pneumonia (%)	25,222 (9.0)	9394 (7.0)	15,828 (10.8)	<0.0001
Hospitalization, n (%)	16,437 (5.9)	6258 (4.7)	10,179 (6.9)	<0.0001
Pneumonia (%)	5154 (1.8)	2079 (1.6)	3075 (2.1)	<0.0001
Death (%)	7975 (2.8)	2721 (2.0)	5254 (3.6)	<0.0001

SD = standard deviation; NSAID = non-steroidal anti-inflammatory drug, * On the day the patient was diagnosed with COVID-19 infection, ^†^ Intensity defined as the number of treatment episodes in the previous 2 years before COVID-19 infection date, ^‡^ Event registered after COVID-19 infection date.

**Table 2 antibiotics-10-01364-t002:** Marginal structural model of the adjusted association between exposure to antibiotics among patients with COVID-19 infection and the composite severity endpoint, combining death, hospitalization and/or pneumonia related to COVID-19.

	Patients Diagnosed with Non-Severe COVID-19 (n: 255,457)	Patients Diagnosed with COVID-19 with the Composite Severity Endpoint (n: 25,222)	Univariable		Multivariable *	
			OR (95% CI)	*p*-Value	OR (95% CI)	*p*-Value
Exposed to antibiotics	130,828 (51.2)	15,828 (62.8)	1.50 (1.40–1.60)	<0.0001	1.12 (1.04–1.21)	0.0022
High exposure (≥5 antibiotics) ^†^	12,395 (4.9)	2520 (10.0)	1.81 (1.73–1.90)	<0.0001	1.19 (1.14–1.26)	<0.0001
Recent exposure (<2 months) ^†^	51,069 (39.0)	8107 (51.2)	1.64 (1.59–1.70)	<0.0001	1.41 (1.36–1.46)	<0.0001
Past exposure (≥2 months) ^†^	79,759 (61.0)	7721 (48.8)	1.35 (1.24–1.46)	<0.0001	1.03 (0.95–1.13)	0.4722
Exposed to HPCIAs ^†^	40,891 (31.3)	6586 (41.6)	1.57 (1.51–1.62)	<0.0001	1.35 (1.30–1.40)	<0.0001
**Covariables**						
Smoking	100,143 (39.2)	10,638 (42.2)	1.22 (1.14–1.30)	<0.0001		
Obesity	65,194 (25.5)	10,545 (41.8)	1.58 (1.48–1.69)	<0.0001		
Ischemic heart disease	5453 (2.1)	2253 (8.9)	2.57 (2.31–2.86)	<0.0001	1.26 (1.11–1.42)	0.0003
Diabetes mellitus	17,704 (6.9)	5900 (23.4)	2.58 (2.39–2.80)	<0.0001	1.53 (1.41–1.66)	<0.0001
High blood pressure	45,614 (17.9)	12,159 (48.2)	2.92 (2.73–3.13)	<0.0001	1.89 (1.75–2.04)	<0.0001
Heart failure	3257 (1.3)	1999 (7.9)	3.44 (3.05–3.87)	<0.0001	1.61 (1.42–1.84)	<0.0001
Chronic kidney disease	7864 (3.1)	4051 (16.1)	3.23 (2.96–3.52)	<0.0001	1.72 (1.56–1.89)	<0.0001
Respiratory disease	39,608 (15.5)	6323 (25.1)	1.65 (1.53–1.77)	<0.0001	1.18 (1.09–1.28)	<0.0001
Thromboembolism	628 (0.2)	268 (1.1)	2.65 (1.93–3.64)	<0.0001	1.61 (1.15–2.26)	0.0056
Use of NSAIDs	59,216 (23.2)	7548 (29.9)	1.21 (1.13–1.30)	<0.0001	1.30 (1.20–1.42)	<0.0001
Antithrombotic medication	11,290 (4.4)	3860 (15.3)	2.25 (2.07–2.45)	<0.0001	1.48 (1.33–1.65)	<0.0001
Use of corticosteroids	13,122 (5.1)	1541 (6.1)	1.10 (0.97–1.25)	0.1513	1.14 (0.99–1.31)	0.0773
Low molecular weight heparin	1147 (0.4)	975 (3.9)	4.62 (3.70–5.78)	<0.0001	4.54 (3.59–5.76)	<0.0001

HPCIA = Highest Priority Critically Important Antimicrobial; NSAID = non-steroidal anti-inflammatory drug; OR = Odds ratio; CI = confidence interval, * Weighted for age, sex, socioeconomic status, and adjusted for the variables included in the table: smoking status, body mass index, comorbidities of interest, concomitant medication. Final set of predictors selected by means of AIC-based stepwise algorithm, ^†^ Compared to all other individuals exposed to antibiotics.

## Data Availability

In accordance with current European and national law, the data used in this study are only available to the researchers participating in this study. Thus, we are not allowed to distribute or make the data publicly available to other parties. However, researchers from public institutions can request data from SIDIAP if they comply with certain requirements. Further information is available online (https://www.sidiap.org/index.php/menu-solicitudes-en/application-proccedure, accessed on 4 November 2021) or by contacting Anna Moleras (amoleras@idiapjgol.org). The analytic code used in this study is freely available at https://github.com/SIDIAP/MultiStateCOVID-19, accessed on 4 November 2021.

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
