# Peer review of "Correlation between Previous Antibiotic Exposure and COVID-19 Severity. A Population-Based Cohort Study"

_antibiotics, 2021, doi:10.3390/antibiotics10111364_

Round 1
Reviewer 1 Report
The study proposed investigated the influence of the previous antibiotic exposure, in particular to highest priority critically important antimicrobials (HPCIA), and the risk of increased COVID-19 disease severity. They included in the study 280.674 patients with confirmed or suspected COVID-19. Of this a total of 25,222 patients presented the primary composite outcome of death, hospitalization and/or pneumonia related to COVID-19 infection. This percentage of disease severity was significantly higher among patients exposed to antibiotics (n: 15,828, 10.8%) compared to patients not exposed (n: 9,394, 7%).
The methodology could be improved
Statistics appear to conform to the predicted outcome.
the tables and graphs are well drafted and explanatory.
The results reflect the conclusions.
Question 1: Given that the antibiotics induce resistances, is it possible to carry out an analysis on the infection (except pneumonia) in patients included in different groups?
Question 2: Given the resistances induced by antibiotics, is it possible to carry out an analysis on the antibiotics prescription (classes) in patients with infection?
Question 3: Is it possible to perform an analysis between two groups considering patients with diagnosis of resistant bacterial infection?
Question 4: Given the well known risk of bad outcome in patients with COPD, high blood pressure, obesity, etc, can you perform an analysis considering only patients with past antibiotics prescription?
Author Response
The study proposed investigated the influence of the previous antibiotic exposure, in particular to highest priority critically important antimicrobials (HPCIA), and the risk of increased COVID-19 disease severity. They included in the study 280.674 patients with confirmed or suspected COVID-19. Of this a total of 25,222 patients presented the primary composite outcome of death, hospitalization and/or pneumonia related to COVID-19 infection. This percentage of disease severity was significantly higher among patients exposed to antibiotics (n: 15,828, 10.8%) compared to patients not exposed (n: 9,394, 7%).
The methodology could be improved. Statistics appear to conform to the predicted outcome. The tables and graphs are well drafted and explanatory. The results reflect the conclusions.
Question 1: Given that the antibiotics induce resistances, is it possible to carry out an analysis on the infection (except pneumonia) in patients included in different groups?
Response: As mentioned in the discussion, this study has some limitations inherent to electronic database studies. Codes for diagnostic classification of chronic conditions are generally reliable, but when it comes to acute conditions the database is incomplete. Only few types of acute infections do have a valid registration, e.g. COVID-19 infection, but most other acute infections are incompletely registered. Therefore, we decided not to perform the analysis proposed by the referee as it may lead to misinterpretation.
Question 2: Given the resistances induced by antibiotics, is it possible to carry out an analysis on the antibiotics prescription (classes) in patients with infection?
Response: We have added this information and included an additional table, see below. This table could be included in the paper or added as supplementary material.
‘Among individuals who were exposed to antibiotics, those who presented the most severe cases of COVID-19 infection were the patients exposed to quinolones (OR, 1.49; 95% CI: 1.42-1.56) and cephalosporins (OR, 1.45; 95% CI: 1.36-1.55) (Appendix table 2).’
TABLE S2. Marginal structural model of the adjusted association between exposure to the most common families of antibacterial agents among patients with COVID-19 infection and the composite severity endpoint, combining death, hospitalization and/or pneumonia related to COVID-19 infection.
|
|
Patients exposed to antibiotics and diagnosed with non-severe COVID-19 (n:130,828) |
Patients exposed to antibiotics and diagnosed with COVID-19 with the composite severity endpoint (n:15,828) |
Univariable |
Multivariable* |
||
|
OR (95% CI) |
P-value |
OR (95% CI) |
P-value |
|||
|
J01C - Penicillins |
71,065 (54.3) |
6,680 (42.2) |
0.61 (0.59 - 0.63) |
<0.0001 |
0.72 (0.69 -0.74) |
<0.0001 |
|
J01D - Cephalosporins |
6,270 (4.8) |
1,261 (8.0) |
1.72 (1.61 - 1.83) |
<0.0001 |
1.45 (1.36 - 1.55) |
<0.0001 |
|
J01F - Macrolides, lincosamides and streptogramins |
23,985 (18.3) |
2,708 (17.1) |
0.92 (0.88 - 0.96) |
0.0001 |
0.99 (0.94 - 1.04) |
0.6446 |
|
J01M - Quinolones |
14,402 (11.0) |
3,268 (20.6) |
2.10 (2.02 - 2.19) |
<0.0001 |
1.49 (1.43 - 1.56) |
<0.0001 |
|
J01X - Other antibacterial agents |
12,448 (9.5) |
1,357 (8.6) |
0.89 (0.84 - 0.95) |
0.0001 |
0.91 (0.85 - 0.96) |
0.0017 |
Question 3: Is it possible to perform an analysis between two groups considering patients with diagnosis of resistant bacterial infection?
Response: Interesting question; however, the available database does give access to data for such an analysis.
Question 4: Given the well known risk of bad outcome in patients with COPD, high blood pressure, obesity, etc, can you perform an analysis considering only patients with past antibiotics prescription?
Response: As suggested by the reviewer we have included past antibiotic prescription (previous exposure (>= 2 months) in table 2 and have added this sentence to the Results section: ‘Patients with past antibiotic exposure (≥ 2 months) had no increased risk of COVID-19 severity.’

Reviewer 2 Report
Τhe authors report the results of a population-based observational cohort study examining whether previous antibiotic exposure could affect COVID-19 severity and outcome. The concept is interesting to remind us about the serious consequences of antimicrobial overuse in gut microbiota.
Two minor issues:
- Introduction: Is there any evidence that antimicrobial overuse affects the outcome of other viral infections by alterations of gut flora? Please comment on that as this is important aspect of your hypothesis
- Conclusions: It would be useful to add a comment about antimicrobial stewarship that should not be neglected during the COVID-19 pandemic
Author Response
Τhe authors report the results of a population-based observational cohort study examining whether previous antibiotic exposure could affect COVID-19 severity and outcome. The concept is interesting to remind us about the serious consequences of antimicrobial overuse in gut microbiota.
Two minor issues:
Introduction: Is there any evidence that antimicrobial overuse affects the outcome of other viral infections by alterations of gut flora? Please comment on that as this is important aspect of your hypothesis
Response: We have added a new paragraph to the Introduction section and three new references to better back up our hypothesis: We have also deleted the first sentence of the last paragraph as this is repeated in the first sentence of the Discussion section:
The interplay between bacteria, viruses and host physiology is complex, and we still have much to learn. Despite this, an increasing body of evidence is beginning to reveal how antibiotic exposure appears to impair antiviral immunity. Some papers carried out in rodents have found how antibiotic exposure among pregnant mice causes substantial alterations to theirs and their offspring’s gastrointestinal microbiota and increased mortality following viral infection [14]. The evidence in humans, is however, scarce. In a recent study, Zhou et al showed that perinatal antibiotic exposure for preventing group B streptococcus infection in newborns induced microbiota dysbiosis in maternal vaginal and neonatal gut environments and was associated with an increased risk of the occurrence of early-onset sepsis among the latter [15]. Another study showed how dysbiosis within the vaginal microbiota caused by oral antibiotic treatment results in severe impairment of antiviral protection against herpes simplex virus type 2 infection [16]. Probiotics, however, would have an opposite role and some selected probiotics have been reported to increase natural killer cell activity and cytotoxic activity [17].
- Gonzalez-Perez, G.; Hicks, A.L.; Tekieli, T.M.; Radens, C.M.; Williams, B.L.; Lamousé-Smith, E.S.N. Maternal antibiotic treatment impacts development of the neonatal intestinal microbiome and antiviral immunity. J. Immunol. 2016, 196, 3768–3779.
- Zhou, P.; Zhou, Y.; Liu, B.; Jin, Z.; Zhuang, X.; Dai, W., Yang, Z.; Feng, Z.; Zhou, Q.; Liu, Y.; Xu, X.; Zhanga, L. Perinatal antibiotic exposure affects the transmission between maternal and neonatal microbiota and is associated with early-onset sepsis. mSphere. 2020, 5, e00984-19.
- Oh, J.E.; Kim, B.C.; Chang, D.H.; Kwon, M.; Young Lee, S.; Kang, D.; Kim, J.Y.; Hwang, I.; Yu, J.W.; Nakae, S.; Lee, H.K. Dysbiosis-induced IL-33 contributes to impaired antiviral immunity in the genital mucosa. Proc. Natl. Acad. Sci. U. S. A. 2016, 113, E762–E771.
- Harper, A.; Vijayakumar, V.; Ouwehand, A.C.; Ter Haar, J.; Obis, D.; Espadaler, J.; Binda, S.; Desiraju, S.; Day, R. Viral infections, the microbiome, and probiotics. Front. Cell. Infect. Microbiol. 2021, 12, 10:596166.
Conclusions: It would be useful to add a comment about antimicrobial stewardship that should not be neglected during the COVID-19 pandemic
Response: We agree with the referee and have added another paragraph at the end of the Discussion section and 3 new references, as follows:
During 2020, the COVID-19 pandemic, an unprecedented global public health crisis, profoundly altered the context for antimicrobial stewardship, mainly in primary care. After the indiscriminate antibiotic prescribing during the first wave of the pandemic, the WHO issued a guidance on the clinical management of COVID-19 recommending that antibiotics should not been prescribed for the prevention and treatment of mild COVID-19 infection. The NICE guideline subsequently suggested that COVID-19 patients presenting with pneumonia symptoms are more likely to have a viral origin than a community-acquired bacterial pneumonia, and thus antibiotic prescriptions should be offered only where bacterial infections are suspected [22]. A subsequent meta-analysis confirmed this as only 3.5% of all COVID-19 patients present with bacterial co-infection [23]. Therefore, there is no evidence that common infections such as respiratory and urinary tract infections should be managed differently during the pandemic since inappropriate use of antibiotics to treat viral infections and indiscriminate use of broad-spectrum antibacterials may lead to resistance [24]. In addition, this practice is associated with deleterious effects on the microbiome and may pose a risk for subsequent severe viral infections as shown in the current study. The ongoing overall crisis of antimicrobial resistance must not be neglected and advocacy for antimicrobial stewardship must continue during the pandemic and the post-pandemic era.
- National Institute for Health and Care Excellence (2020) COVID-19 rapid guideline: managing suspected or confirmed pneumonia in adults in the community NG165 (NICE, London). Available online: https://www.nice.org.uk/guidance/ng165 (accessed on 2 October 2021).
- Langford, B.J.; So, M.; Raybardhan, S.; Leung, V.; Westwood, D.; MacFadden, D.R.; Soucy, J.P.R.; Daneman, N. Bacterial co-infection and secondary infection in patients with COVID-19: a living rapid review and meta-analysis. Clin. Microbiol. Infect. 2020, 26, 1622-1629.
- Khor, W.P.; Olaoye, O.; D’Arcy, N.; Krochow, E.M.; Elshanawy, R.A.; Rutter, V.; Ashiru-Oredope, D. The need for ongoing antimicrobial stewardship during the COVID-19 pandemic and actionable recommendations. Antibiotics. 2020, 9; 904.
Round 2
Reviewer 1 Report
the authors answered my questions extensively according to their possibilities.
Author Response
Thank you very much for your review.